# Exploration of Bromodomain Proteins as Drug Targets for Niemann–Pick Type C Disease

**DOI:** 10.3390/ijms26125769

**Published:** 2025-06-16

**Authors:** Martina Parente, Amélie Barthelemy, Claudia Tonini, Sara Caputo, Alessandra Sacchi, Stefano Leone, Marco Segatto, Frank W. Pfrieger, Valentina Pallottini

**Affiliations:** 1Department of Science, Section Biomedical Science and Technology, University Roma Tre, Viale Marconi 446, 00146 Rome, Italy; martina.parente@uniroma3.it (M.P.); claudia.tonini@uniroma3.it (C.T.); sara.caputo@uniroma3.it (S.C.); alessandra.sacchi@uniroma3.it (A.S.); stefano.leone@uniroma1.it (S.L.); 2Institut des Neurosciences Cellulaires et Intégratives, Centre National de la Recherche Scientifique, Université de Strasbourg, 8 Allée du Général Rouvillois, 67000 Strasbourg, France; amelie.barthelemy.ae@gmail.com; 3Department of Biosciences and Territory, University of Molise, Contrada Fonte Lappone s.n.c., 86090 Pesche, Italy; marco.segatto@unimol.it; 4Neuroendocrinology Metabolism and Neuropharmacology Unit, IRCSS Fondazione Santa Lucia, Via del Fosso Fiorano 64, 00143 Rome, Italy

**Keywords:** cholesterol, drug therapy, epigenetic regulation, lysosome, rare disease, BET inhibitor

## Abstract

Defects in lysosomal cholesterol handling provoke fatal disorders presenting neurovisceral symptoms with variable onset and life spans. A prime example is Niemann–Pick type C disease (NPCD), where cholesterol export from the endosomal–lysosomal system is impaired due to variants of either NPC intracellular cholesterol transporter 1 (NPC1) or NPC intracellular cholesterol transporter 2 (NPC2). Therapeutic options for NPCD are limited to palliative care and disease-modifying drugs, and there is a need for new treatments. Here, we explored bromodomain and extra-terminal domain (BET) proteins as new drug targets for NPCD using patient-derived skin fibroblasts. Treatment with JQ1, a prototype BET protein inhibitor, raised the level of NPC1 protein, diminished lysosomal expansion and cholesterol accumulation, and induced extracellular release of lysosomal components in a dose-, time-, and patient-dependent manner. Lastly, JQ1 enhanced and reduced cholesterol accumulation induced by pharmacologic inhibition of NPC1 and of histone deacetylase (HDAC) activity, respectively. Taken together, bromodomain proteins should be further explored as therapeutic drug targets for lysosomal diseases like NPCD, and as new components regulating lysosomal function and cholesterol metabolism.

## 1. Introduction

Lysosomal handling of lipids is essential for cellular function [1], and genetic defects in these processes provoke fatal disorders presenting highly variable onset, diverse neurovisceral symptoms, and reduced life spans [2]. A prime example is NPCD, a rare, autosomal-recessive, and pan-ethnic lysosomal disorder presenting progressive and ultimately fatal neurovisceral symptoms [3,4]. Several forms of NPCD are discerned based on the onset of neurologic disease [5,6,7]. Disease incidence is estimated at 1:100,000 [3].

The primary cause of NPCD is specific alleles of the gene *NPC intracellular cholesterol transporter 1* (*NPC1*; OMIM #257220; 95% of cases) [8,9] or the gene *NPC intracellular cholesterol transporter 2* (*NPC2*; OMIM 607625; 5% of cases) [10]. The encoded proteins NPC1 and NPC2, which are ubiquitously expressed, reside in the membrane [11,12] and the lumen of late endosomes [10,13,14], respectively. Phenotypes of mouse models [15], biochemical assays [16,17], and structure analyses [18,19,20,21,22,23] indicate that the two proteins export unesterified cholesterol from the endosomal–lysosomal system, although other functions are discussed. Dysfunction of either protein causes intracellular accumulation of unesterified cholesterol [24,25] and of other lipids [26,27,28,29,30], and impairs lysosomal [29,31,32,33] and mitochondrial function [34], and autophagy [35].

Despite considerable efforts [36], therapeutic options for NPCD are limited to symptomatic treatment and to the disease-modifying drugs N-butyl-deoxynojirimycin (OGT918, Miglustat, Zavesca) [37], arimoclomol (Miplyffa) in combination with Miglustat [38], and N-acetyl-L-leucine (Levacetylleucine) [39].

Here, we explored BET proteins as new drug targets for NPCD. BET proteins control gene expression in a complex manner. They recognize and bind to specific acetyl-lysine patterns on histones, control chromatin structure, and interact with transcription factors to promote or repress transcriptional programs [40,41]. They are considered therapeutic drug targets [42] for different types of cancer [43] and other pathologic conditions [44,45] including cachexia [46], Duchenne muscular dystrophy [47], Fabry disease [48], and retinal inflammation [49]. Recently, we discovered that JQ1, a well-characterized competitive inhibitor of BET proteins [50], enhances the protein content of NPC1 in cultured cells [51]. This effect is in line with evidence that BET proteins control lysosome- and autophagy-related genes in various cell types and disease conditions [47,52,53,54,55,56,57,58,59]. Thus, inhibition of BET proteins may have therapeutic potential in NPCD because disease severity seems to correlate with cellular levels of NPC1 protein [60,61,62] and because some pathogenic variants of NPC1 are functional, but degraded due to misfolding [63,64,65,66,67,68].

Using patient-derived skin fibroblasts as a preclinical model for in vitro drug tests [69], we observed that inhibition of BET proteins by JQ1 increased protein levels of NPC1 and affected pathologic changes in a dose-, time-, and patient-dependent manner. The drug induced an enhancement of lysosomal release, followed by an increase in cholesterol accumulation, and a subsequent decrease after long-term treatment. Effects of JQ1 on cholesterol accumulation were abolished by an NPC1 inhibitor, and varied across fibroblasts from distinct patients. Together, our results support further exploration of BET proteins as new therapeutic targets for NPCD and as new components regulating lysosomal function and cholesterol metabolism.

## 2. Results

We explored BET proteins as new therapeutic drug targets for NPCD using the membrane-permeant inhibitor JQ1 and primary cultures of dermal fibroblasts from NPCD patients and a healthy donor.

### 2.1. Effects of JQ1 on Viability, Protein Levels of NPC1, and Its Distribution in Human Skin Fibroblasts

First, we tested whether the drug affects the number and viability of fibroblasts. As shown in Figure 1A, patient-derived fibroblasts carrying the I1061T variant attained lower numbers and showed a lower fraction of propidium iodide-positive (dead) cells compared to fibroblasts from the healthy donor. JQ1 inhibited cell growth and enhanced the percentage of dead cells compared to vehicle (dimethylsulfoxide; DMSO) in a dose- and time-dependent manner, regardless of the genotype (Figure 1B). Anti-proliferative and toxic effects of JQ1 at high doses were reported in dermal fibroblasts [70] and in immortalized cell lines [50,71].

Next, we tested how JQ1 affects NPC1 protein levels in human skin fibroblasts. These experiments followed up on our previous observation that the drug modifies components mediating lipid homeostasis, including NPC1, in a hepatocarcinoma cell line [51]. Immunoblotting revealed lower levels of the I1061T variant compared to the normal version of NPC1 in fibroblasts (Figure 2A), in line with previous studies [63,72,73]. JQ1 enhanced protein levels of NPC1 in a concentration-dependent manner compared to vehicle-treated (JQ1 concentration zero) cultures independently from the genotype, with robust changes occurring after 72 h of treatment (Figure 2B). The fraction of endoglycosidase H (EndoH)-resistant NPC1 protein was unaffected by JQ1 regardless of the variant (Figure 2C), indicating that JQ1 does not modify protein glycosylation under our experimental conditions.

Next, we asked whether JQ1-induced NPC1 protein reaches the endosomal–lysosomal system using double immunocytochemical staining with antibodies against NPC1 and against the endosomal–lysosomal marker LAMP2 (Figure 3A). Quantitative analyses of regions of interest (ROIs) confirmed lower levels of NPC1 and of its colocalization with LAMP2 in patient-derived fibroblasts compared to healthy donor cells (Figure 3B). Treatment of patient-derived cells with JQ1 for 48 or 72 h did not enhance the density of LAMP2 or NPC1-positive clusters (Figure 3A,C) and only moderately affected the colocalization of NPC1 and LAMP2 (Figure 3C).

### 2.2. Effects of JQ1 on Lysosomes in Patient-Derived Skin Fibroblasts

We tested next whether JQ1 affects the expansion of the acidic (lysosomal) compartment, a cellular hallmark of NPC1 deficiency, using lysotracker, a pH-sensitive fluorescent dye [74,75,76] (Figure 4A–D). Flow cytometry of lysotracker-labeled fibroblasts revealed a higher cell complexity and increased fluorescence signal in cells carrying the disease-causing variant of NPC1 compared to cells from a healthy donor (Figure 4A–C). Treatment of patient-derived cells with JQ1 reduced the lysotracker signal in a dose-dependent manner regardless of the treatment duration (Figure 4D). This effect may have been due to exocytotic release of lysosomal content [77]. To address this, we used two complementary assays [77,78,79]. First, immunocytochemical staining of LAMP2 in non-permeabilized cells revealed the presence of LAMP2 on the cell surface following fusion of the lysosome with the plasma membrane. Second, detection of hexosaminidase activity in the culture medium revealed the cellular release of lysosomal enzymes (Figure 4E–I). The assays showed enhanced basal LAMP2 surface expression (Figure 4E,F) and extracellular hexosaminidase activity (Figure 4H) in patient-derived fibroblasts compared to fibroblasts from a healthy donor, in agreement with a previous study [80]. Treatment with JQ1 further increased both the LAMP2 surface expression (Figure 4G) and extracellular hexosaminidase activity in a time- and concentration-dependent manner (Figure 4I), suggesting that the drug induces release of lysosomal material.

### 2.3. Effects of JQ1 on Cholesterol Accumulation in Patient-Derived Skin Fibroblasts

We tested next whether JQ1 affects the accumulation of unesterified cholesterol, another hallmark of NPC1 deficiency, using cytochemical staining with the cholesterol-binding drug filipin [24,81] (Figure 5). Patient-derived fibroblasts showed a higher density of filipin-positive puncta (Figure 5A,B) compared to cells from the healthy donor (Figure 5B). Interestingly, JQ1 increased the density of filipin-positive puncta after 48 h of treatment (Figure 5C) and showed no effects after 72 h (Figure 5C). The increase may have been due to enhanced cholesterol synthesis. However, cellular levels of 3-Hydroxy-3-Methylglutaryl-CoA Reductase (HMGCR), a key enzyme mediating cholesterol synthesis, were unaffected by JQ1 during these treatment periods, as shown by immunoblotting (Appendix A). Closer scrutiny of filipin-stained cells treated for 72 h with JQ1 revealed that the drug induces a dual effect in a given cell population, with the majority showing decreased and 30% of cells showing increased filipin densities (Figure 5D). These divergent effects explained the lack of significance and suggested a time-dependent switch of the JQ1 effect. We treated cells with JQ1 for 168 h and found that long-term treatment reduced the density of filipin-positive puncta in patient-derived fibroblasts (Figure 5C,D), indicating time- and dose-dependent effects of the drug on cholesterol accumulation.

### 2.4. Effects of JQ1 on Skin Fibroblasts from Different NPCD Patients

We next tested how JQ1 affects NPC1 levels and cholesterol accumulation in skin fibroblasts from four patients bearing different variants and polymorphisms. As shown in Figure 6, these lines showed different basal levels of NPC1 protein and different degrees of cholesterol accumulation under untreated conditions. Treatment with JQ1 for 72 h raised the protein levels of NPC1 in a dose-dependent manner in all but one fibroblast line (GM17920). With respect to cholesterol accumulation, the effects of JQ1 differed markedly between patient-derived lines. The spectrum ranged from a robust decrease already within 72 h in line GM03123 to an apparent lack of response in line GM17920, which showed no increase in NPC1 protein, and in line GM17921 despite a strong increase in NPC1 protein. In the other lines, JQ1 reduced cholesterol accumulation after 168 h of treatment (Figure 6D). Notably, in all lines tested, JQ1 showed dual effects, increasing and decreasing puncta density in subsets of fibroblasts from the same culture preparation (Figure 6E) as shown in GM18453 (Figure 5D). These results indicated that JQ1 raises NPC1 levels in most patient-derived lines. With respect to cholesterol accumulation, subsets of cells responded to JQ1 with increased and decreased densities in a time- and dose-dependent manner, with a net reduction occurring in 3 out of 4 lines after prolonged treatment.

### 2.5. Effects of JQ1 on Cholesterol Accumulation in Patient-Derived Fibroblasts in the Presence of an NPC1 Inhibitor

Our findings raised the question of whether the reduction in cholesterol accumulation by JQ1 depends on NPC1 activity. To address this point, we applied JQ1 in the presence or absence of 3-beta-[2-(diethylamine)ethoxy]androst-5-en-17-one (U18666A or U18), an inhibitor of NPC1 [82], using the GM03123 line (Figure 6D). As shown in Figure 7A,B, treatment with U18 enhanced the intensity of filipin fluorescence in a dose-dependent manner, indicating residual NPC1 activity in these cells. Co-treatment with JQ1 further enhanced the staining intensity at each U18 concentration tested (Figure 7). These results suggested that JQ1 reduces cholesterol accumulation in an NPC1-dependent manner. JQ1 may also act through NPC1-independent pathways that are insufficient to overcome U18-induced cholesterol accumulation.

### 2.6. Effects of JQ1 on HDAC Inhibitor-Mediated Reduction in Cholesterol Accumulation in Patient-Derived Fibroblasts

Previous studies revealed that pharmacologic inhibition of HDACs reduces cholesterol accumulation in mouse neural stem cells [83] and human fibroblasts [72,84,85] carrying NPC1 variants. Therefore, we tested whether inhibition of BET proteins acting as histone acetylation readers modifies these effects. In line with previous reports, the HDAC inhibitor suberoylanilide hydroxamic acid (Vorinostat; SAHA) reduced the density of filipin-positive puncta in a dose-dependent manner (Figure 7C,D). Interestingly, similar to JQ1 (Figure 5D), SAHA also induced dual effects with fibroblasts in the same culture well, showing enhanced and reduced puncta densities (Figure 7E). The addition of JQ1 further reduced the density of puncta, but the effect did not reach statistical significance (two-way ANOVA; Figure 7D,E).

## 3. Discussion

Here, we report effects of BET protein inhibition in a cell culture model of NPCD that encourage further exploration of this approach. We found that a prototypical BET protein inhibitor enhanced the cellular level of NPC1 protein, diminished lysosomal expansion and cholesterol accumulation, and induced the release of lysosomal components in a time- and dose-dependent manner. The effects of JQ1 on cholesterol levels depended on NPC1 activity, and the enhancement of protein levels occurred in most of the patient lines tested, but the extent of cholesterol reduction varied in a line-dependent manner.

Our finding that BET protein inhibition enhances protein levels of NPC1 is in line with previous reports that the promoter region of *NPC1* is associated with acetylated histones [86] and that de-acetylation increases gene expression [84]. BET proteins may read these patterns and repress NPC1 production. Other means to enhance cellular NPC1 protein levels in vitro are the inhibition of HDACs [65,68,72,87,88,89], enhanced chaperone activity [63,73,90,91,92,93,94,95,96,97], and reduced protein degradation [63,64,90,92]. These manipulations enhanced the presence of NPC1 variants in the endosomal–lysosomal system [63,64,68,73,89,90,91,94,95,96], and they reduced the intracellular accumulation of unesterified cholesterol [63,64,65,68,72,73,84,85,88,89,90,91,95,96]. There is evidence for synergistic effects of BET protein and HDAC inhibitors in transcription regulation [98] and tumor therapy [99,100], but in NPCD patient-derived fibroblasts, JQ1 did not significantly enhance the SAHA-induced reduction in cholesterol accumulation.

Our observation that the responses to JQ1 are patient-dependent is in line with patient-specific effects of HDAC inhibitors [68,72,89]. The outcome probably depends on the specific activity of NPC1 variants generated from each patient’s allele combinations [63,68,89]. Evidently, so-far-unknown genetic or epigenetic modifiers [101,102,103] may further impact the outcome of JQ1 treatment. The effects of JQ1 seemed to depend on NPC1 activity since JQ1 failed to reduce cholesterol accumulation after pharmacologic inhibition of NPC1 by U18. However, this finding does not exclude that JQ1 also acts through NPC1-independent pathways.

Our observation that at 72 h treatment, JQ1 increased and decreased cholesterol accumulation in cells may explain why JQ1 did not affect cholesterol accumulation in a recent fibroblast-based high-throughput drug screen for NPCD [104]. The time-dependent effects of JQ1 may be due to sequential activation of distinct processes or subtype-specific responses [105]. An initial increase in cholesterol accumulation may be due to enhanced levels of lysosomal components as reported previously [52] and an insufficient integration of NPC1 in the endosomal–lysosomal system. During this period, JQ1 also reduced the lysosomal volume as indicated by lysotracker staining. This effect may have been caused by the immediate release of lysosomal content to the extracellular space, as indicated by two independent assays. Lysosomal exocytosis reduces the extent of cholesterol accumulation due to NPC1 dysfunction, as shown previously in cell lines [78,80,106,107,108], patient-derived fibroblasts [80,109,110,111,112,113], primary retinal neurons [79,114], and in NPC1-deficient mice [115]. The net decrease in cholesterol accumulation after prolonged JQ1 treatment may be caused by additional processes that are affected by BET protein inhibition [116], including a reduction in cholesterol biosynthesis [51,117] and of intracellular lipid levels [51], and an increase in apolipoprotein A [118], which has been explored as therapeutic agent for NPCD [119]. In general, BET protein inhibition affects many transcriptional programs through interactions with acetylated lysines on histones, including a site showing epigenetic marks in NPCD [120]. Notably, a lack of effect of BET protein inhibition in some patient lines does not exclude a therapeutic effect of this approach. Miglustat and arimoclomol, which are approved for the treatment of NPCD, did not reduce cholesterol accumulation in cellular models in vitro [95,121,122], and cerebellar Purkinje cells in NPC1-deficient cats showed cholesterol accumulation following miglustat treatment [123].

Taken together, our results reveal that BET proteins regulate NPC1 levels and thereby impact lysosomal function and cholesterol homeostasis depending on specific protein variants and the genetic background. Our results encourage further studies to evaluate their potential as a therapeutic drug target for NPC disease and their contribution to cholesterol homeostasis and lysosomal function in differentiated cells.

## 4. Materials and Methods

### 4.1. Cell Culture and Drug Treatment

Human dermal fibroblasts used in this study were obtained from the NIGMS Human Genetic Cell Repository [Coriell Institute for Medical Research, Camden, NJ, USA]. Most experiments were performed using fibroblasts from a NPCD patient homozygous for a frequent pathogenic allele [GM18453, https://catalog.coriell.org/0/Sections/Search/Sample_Detail.aspx?Ref=GM18453 (accessed on 10 June 2025) male, p.Ile1061Thr p.Ile1061Thr] [124] and from a sex- and age-matched healthy donor [GM05659 https://catalog.coriell.org/0/Sections/Search/Sample_Detail.aspx?Ref=GM05659 (accessed on 10 June 2025): male, 14 months old]. Selected experiments were performed using fibroblasts from heterozygous patients carrying different allele combinations [GM00110 https://catalog.coriell.org/0/Sections/Search/Sample_Detail.aspx?Ref=GM00110 (accessed on 10 June 2025): male, 9 years, p.Pro237Ser p.Phe740_Ser741del [60]; GM03123 https://catalog.coriell.org/0/Sections/Search/Sample_Detail.aspx?Ref=GM03123 (accessed on 10 June 2025): female, 9 years, p.Pro237Ser p.Ile1061Thr [60,125]; GM17920 https://catalog.coriell.org/0/Sections/Search/Sample_Detail.aspx?Ref=GM17920 (accessed on 10 June 2025): female, p.Pro401Thr p.Ile1061Thr [94]; GM17921 https://catalog.coriell.org/0/Sections/Search/Sample_Detail.aspx?Ref=GM17921 (accessed on 10 June 2025): male, 5 years, p.Pro433Leu p.Ile1061Thr [126]]. Cells were cultured in Dulbecco’s modified Eagle medium (DMEM) containing high glucose supplemented with 5% fetal bovine serum, 1% L-glutamine, 1% sodium pyruvate, 1% non-essential amino acids, and 1% penicillin/streptomycin (all Sigma-Aldrich/Merck, Milano, Italy) at 37 °C and 5% CO_2_. All experiments were performed at 60–70% cell confluency and maximally 20 passages. The drugs (+)-JQ1 (#SML1524, Sigma-Aldrich/Merck), U18 [#S9669; Selleckchem, Houston, TX, USA] and SAHA (#SML0061, Sigma-Aldrich/Merck) were added to primary cultures at indicated concentrations after dilution from respective stock solutions (JQ1: 3 mM; SAHA: 1 mM in DMSO; U18: 5 µg/mL in ethanol). Control cultures run in parallel were treated with vehicle (0.1% DMSO or 0.1% ethanol or both in DMEM) for the indicated times.

### 4.2. Cell Number and Viability

Fibroblasts were cultured in 24-well plates (#833922, Sarstedt, Nürmbrecht, Germany) at 15,000 cells/well and treated as indicated. Cells were detached with trypsine/EDTA (0.05%), resuspended in medium, stained with propidium iodide (2 µg/mL; Sigma-Aldrich/Merck) to label dead cells, and subjected to flow cytometry (CytoFlex, Beckman Coulter, Brea, CA, USA) counting all cells and propidium iodide-positive cells (excitation at 488 nm, emission 585/42 band pass filter).

### 4.3. Lysate Preparation and Immunoblotting

Fibroblasts were cultured in 6-well plates at 150,000 cells/well and treated as indicated. Cells were lysed in homogenization buffer (sucrose 0.1 M, KCl 0.05 M, KH_2_PO_4_ 0.04 M, EDTA 0.04 M, pH 7.4, with proteinase (1:1000; #P8340; Sigma-Aldrich/Merck) and phosphatase inhibitor cocktails (1:400; #P0044; Sigma-Aldrich/Merck) by sonication [VCX 130 PB, Sonics Materials, Newtown, CT, USA] on ice for 20 sec. Then, samples were spun down at 13,000 rpm for 10 min at 4 °C to remove cell debris. Protein concentrations were assessed by the Bradford method (Sigma-Aldrich/Merci) following the manufacturer’s instructions. For immunoblotting, samples were diluted with Laemmli buffer, boiled for 5 min, and subjected to SDS-PAGE (40 µg of protein/lane). Proteins were transferred to nitrocellulose membranes [Trans-Blot Turbo Transfer System; Bio-Rad Laboratories, Hercules, CA, USA]. Membranes were blocked with fat-free milk (5% in Tris-buffered saline 0.138 M NaCl, 0.027 M KCl, 0.025 M Tris-HCl, and 0.05% Tween-20, pH 6.8) for 1 h at room temperature, exposed to antibodies against NPC1 (1:1000; #NB400-148, Novus Biologicals / Bio-Techne S.A.S. Noyal Châtillon sur Seiche, France) or vinculin as loading control (1:40,000; #V9131, Sigma-Aldrich/Merck) or against HMGCR (1:2000, #ab174830; Abcam, Cambridge, UK) and against tubulin (TUB) as loading control (1:40,000; #T6074 Sigma-Aldrich/Merck) overnight at 4 °C followed by corresponding horseradish peroxidase-conjugated secondary IgG antibodies (Bio-Rad Laboratories) for 1 h at room temperature. Chemiluminescence was visualized using the ChemiDoc MP system (Bio-Rad Laboratories) and analyzed by ImageJ software [Version 8; National Institutes of Health, Bethesda, MD, USA].

### 4.4. Endoglycosidase H Assay

Fibroblasts were cultured in 6-well plates and treated as indicated. Cells were lysed in homogenization buffer (sucrose 0.1 M, KCl 0.05 M, KH2PO4 0.04 M, EDTA 0.04 M, pH 7.4) by sonication (VCX 130 PB, Sonics Materials) at 4 °C for 20 s, and centrifuged at 12,000 rpm for 10 min at 4 °C to yield total lysate. The endoglycosidase H (EndoH) assay (V4875, Promega Italia; Milano, Italy) was performed following the manufacturer’s instructions. The samples contained EndoH reaction buffer, water, and EndoH enzyme. As a negative control, the EndoH enzyme was replaced by water. All samples were incubated at 37 °C for 6 h, and the reaction was terminated by adding Laemmli sample buffer before immunoblotting.

### 4.5. Immunocytochemical Staining

Fibroblasts were cultured in 96-well microplates (Black/Clear Flat Bottom Imaging Microplate; #353219, BD Falcon, Thermo Fisher Scientific, St. Leon-Rot, Germany) at 3000 cells/well and treated as indicated. Following treatment, cells were washed three times with phosphate-buffered saline (PBS), and chemically fixed with 4% paraformaldehyde (in PBS) for 15 min at room temperature. Cells were permeabilized (saponin 0.05% in PBS; #84510, Sigma-Aldrich/Merck) for 10 min, and incubated for 45 min with blocking solution (3% bovine serum albumin with 1% goat serum in PBS) and then overnight at 4 °C with primary antibodies (1% bovine serum albumin in PBS) against NPC1 (1:2500; NB400-148, Novus Biologicals) and LAMP2 [1:1000; #sc-18822, Santa-Cruz, Dallas, TX, USA/Clinisciences, Nanterre, France]. After incubation, cells were washed and reacted for 1 h at room temperature with appropriate secondary antibodies (1:1000; goat anti-rabbit secondary antibody Alexa Fluor 546; #A-10040, Thermo Fisher Scientific, St. Leon-Rot, Germany; goat anti-mouse secondary antibody Alexa Fluor 488; #A-11001, ThermoFisher Scientific). Fluorescence was visualized and digitized using an upright microscope (ZEISS Observer 7; Carl Zeiss France S.A.S., Rueil-Malmaison, France/Lordil, Lay-Saint-Chrisophe, France) equipped with a light source (ZEISS Colibri; Carl Zeiss France S.A.S./Lordil), objectives (40× water, N.A. 1.2; 63× oil, N.A. 1.4; Carl Zeiss France S.A.S./Lordil), a module for optical sectioning by structured illumination (ApoTome.2; Carl Zeiss France S.A.S./Lordil) and a digital camera (Hamamatsu ORCA-Flash 4.0; Carl Zeiss France S.A.S./Lordil). Densities of LAMP2- and NPC1-positive puncta were determined in manually outlined regions of interest (ROI) (1–5 per soma) using custom-written LabVIEW (Version 6.0; National Instruments, Austin, TX, USA) routines [79]. Colocalization was estimated based on Pearson’s correlation coefficient of NPC1 and LAMP2 fluorescence intensities in individual puncta detected in ROIs.

### 4.6. Cytochemical Staining

Fibroblasts were cultured in 96-well microplates (black imaging plate; #353219, BD Falcon, Schaffhausen, Switzerland) at 3000 cells/well and treated as indicated. Following treatment, cells were washed three times with PBS, fixed by paraformaldehyde (4% in PBS) for 15 min at room temperature and stained with filipin (50 μg/mL in PBS prepared freshly from a 250-fold ethanolic stock; #F9765, Sigma-Aldrich/Merck) for 2 h at room temperature in the dark. Fluorescence images of stained cells were acquired using an inverted microscope (Axiovert 135TV; Carl Zeiss Microscopy GmbH, Oberkochen, Germany) equipped with a metal halide lamp (10%; Lumen 200, Prior Scientific Instruments GmbH, Jena, Germany), an appropriate excitation/emission filter (XF02-2; Omega Optical, LLC/Laser Components S.A.S., Meudon, France), a 40× objective (oil, N.A. 1.3; Carl Zeiss Microscopy GmbH) and an air-cooled monochrome charge-coupled device camera (Sensicam, PCO Computer Optics, Kehlheim, Germany) controlled by custom-written LabVIEW routines (Version 6.0; National Instruments). Densities of filipin-positive puncta and fluorescence intensities were determined in 10 to 12 images per condition and preparation from manually outlined ROIs (1–5 per cell) using custom-written LabVIEW routines (Version 6.0; National Instruments) [79].

### 4.7. Lysotracker Staining and Cytometry

Fibroblasts were cultured in 6-well plates (#833920, Sarstedt) at 150,000 cells/well and treated as indicated. Before the end of the treatment, cells were incubated with LysoTracker Red DND-99 (1 µM in culture medium; #L7528, Life Technologies/Thermo Fisher Scientific) for 30 min at 37 °C, detached, spun down at 13,000 rpm, and resuspended in 300 μL of medium prior to flow cytometry. Data were acquired using a flow cytometer [Cytoflex-LX; Beckman-Coulter, Brea, CA, USA] and analyzed by specialized software [Kaluza Version 2.1, Beckman-Coulter; FloJo Version 10, Becton-Dickinson, Ashland, OR, USA]. Fold-changes in LysoTracker intensity were calculated as ratios of geometric means of stained/unstained samples as described [74].

### 4.8. Hexosaminidase Activity Assay

Fibroblasts were cultured in 96-well microplates (#833924, Sarstedt) at 4000 cells/well in DMEM without phenol red. The assay was performed similarly as described [79]. Briefly, following treatment, 20 µL of cell culture medium was incubated at 37 °C for 3 h with 20 µL reaction mix containing sodium citrate (10 mM; pH 4.2) and 4-methylumbelliferyl-2-acetamido-2-deoxy-b-D-glucopyranoside (2 mM; #474502, Sigma-Aldrich/Merck). The reaction was stopped by 5 volumes of glycine and Na_2_CO_3_. (0.2 M). Crystal violet (#C0775, Sigma-Aldrich/Merck; 0.05% in H_2_O with 1% paraformaldehyde and 1% methanol) was added to the cell suspension to indicate cell number. The fluorescent product 4-methylumbelliferone and crystal violet were measured in triplicate on a microplate reader (Tecan Spark, Männedorf, Switzerland) using suitable filters (excitation: 365 nm; emission filters: 440 nm; absorbance: 585 nm). Calibration curves were acquired using defined amounts of the fluorescent product 4-methylumbelliferone sodium salt (#M1508, Sigma-Aldrich/Merck). Fluorescence of 4-methylumbelliferone was normalized to crystal violet intensity.

### 4.9. Data Analysis and Visualization

Data analysis and visualization were accomplished with ImageJ software [version 8, National Institutes of Health, Bethesda, MD, USA], and with custom-written routines using the open-source software R (Version 4.0.5; [127] and selected packages (data.table: version 1.14.0, ggplot2: Version 3.3.3). Unless indicated otherwise, results are displayed using bar and whisker plots representing mean and standard deviation, respectively. Statistical tests were performed as indicated. When comparing three or more experimental groups, analysis of variance (one- or two-way ANOVA) was carried out, followed by Tukey’s post hoc test as indicated. Asterisks indicate statistically significant differences based on *p* values (*, *p* < 0.05; **, *p* < 0.01; ***, *p* < 0.001).

## 5. Patents

An Italian patent (No. 102021000015467) has resulted from the work reported in this article.

## Figures and Tables

**Figure 1 ijms-26-05769-f001:**
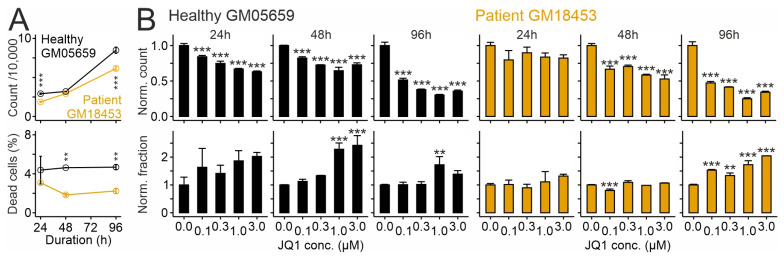
Effects of JQ1 on cell number and viability in cultured human skin fibroblasts. Cell counts (top) and percentages of propidium iodide-positive cells (bottom) in primary cultures of skin fibroblasts from a healthy donor (GM056549; black) and a NPCD patient (GM018453; orange) in untreated cells (**A**) and after treatment with JQ1 or vehicle (DMSO) for indicated periods and concentrations (**B**). Values in (**B**) were normalized to values from vehicle-treated cultures run in parallel. Asterisks indicate statistically significant changes [**, *p* < 0.01; ***, *p* < 0.001; two-way (**A**) and one-way (**B**) analysis of variance (ANOVA) with Tukey’s post hoc test; *n* = 3 independent preparations for each experiment].

**Figure 2 ijms-26-05769-f002:**
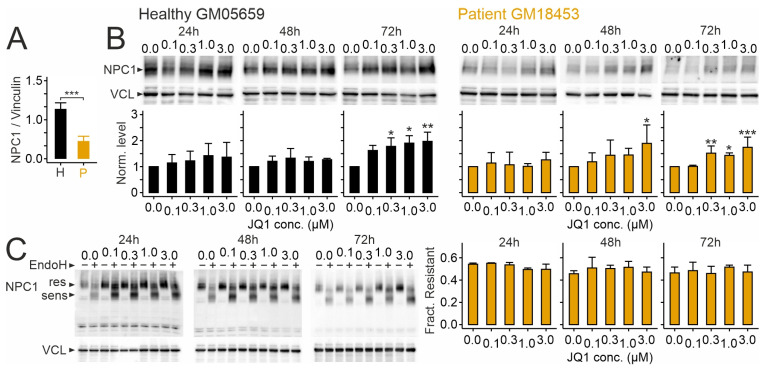
JQ1 enhances NPC1 protein levels in cultured human skin fibroblasts: (**A**,**B**) Levels of NPC1 protein in primary cultures of skin fibroblasts from a healthy donor (H; GM05659; black) and a NPCD patient (P; GM018453; orange) in untreated cells (**A**) and after treatment with JQ1 or vehicle (DMSO) of patient-derived cells for indicated periods and concentrations (**B**). Values in (**A**,**B**) were normalized to levels of vinculin (VCL) protein and to vehicle-treated (JQ1 concentration zero) control cultures, respectively. Asterisks indicate statistically significant changes [*, *p* < 0.05; **, *p* < 0.01; ***, *p* < 0.001; (**A**) patient versus healthy, *t* test, *n* = 4 preparations; (**B**) treatment versus vehicle control, one-way ANOVA with Tukey’s post hoc test; *n* = 3–7 preparations]. (**C**) fraction of EndoH-resistant protein compared to total protein in patient-derived fibroblasts after treatment with JQ1 as indicated (*n* = 3 preparations per treatment). Note the change in protein size following EndoH-mediated glycan removal. Images in (**B**,**C**) show representative immunoblots reacted with antibodies against NPC1 and VCL as a loading control.

**Figure 3 ijms-26-05769-f003:**
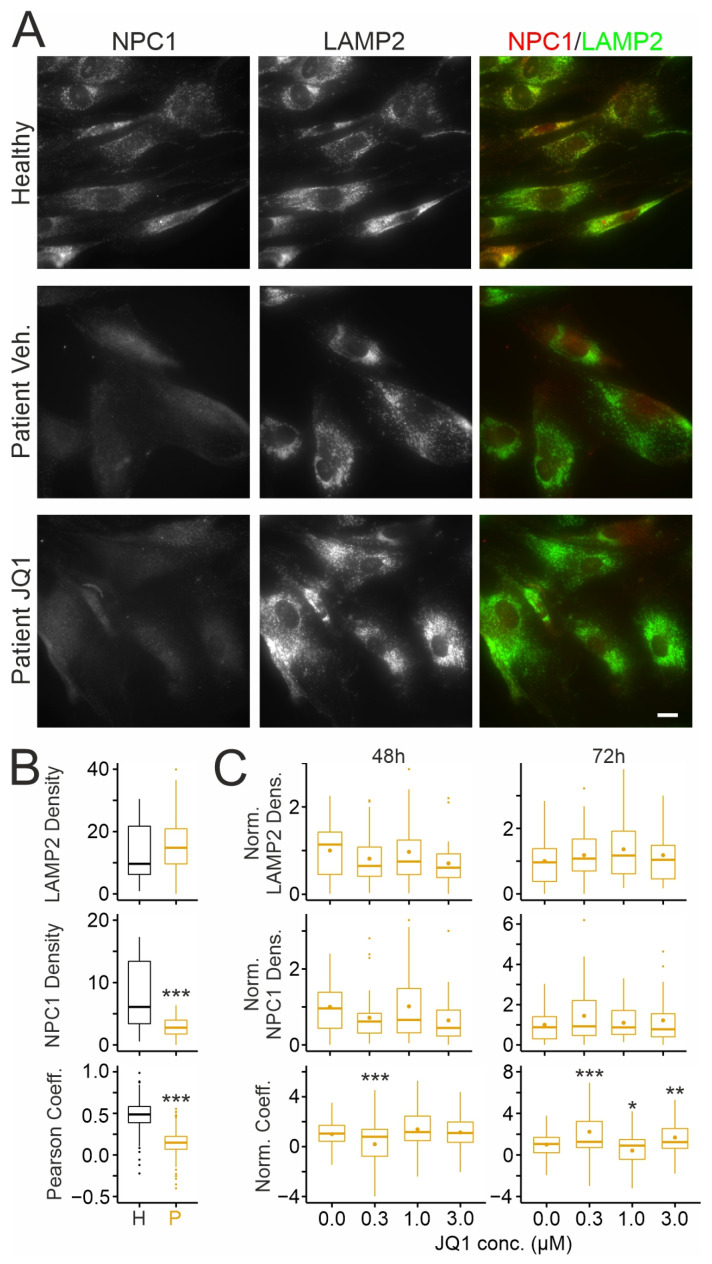
Effects of JQ1 on subcellular distribution of NPC1 in cultured human skin fibroblasts: (**A**) Fluorescence micrographs of cultured fibroblasts from a healthy donor (**top**) (GM056549) and a NPCD patient (GM18453) following treatment for 72 h with vehicle (DMSO) (**middle**) or with JQ1 (3 µM; **bottom**). Cells were fixed and subjected to double immunocytochemical staining for NPC1 (**left**) and LAMP2 (**middle**). False-color micrographs (**right**) represent overlays of NPC1 (red) and LAMP2 fluorescence (green). Scale bar: 20 µm. (**B**,**C**) Boxplots showing the density of LAMP2 (**top**) and of NPC1-positive puncta (**middle**) and Pearson’s correlation coefficients of fluorescence intensities in NPC1- and LAMP2-positive puncta (**bottom**) in somata of patient (P) and healthy donor (H) fibroblasts without treatment (**B**), and in patient fibroblasts following treatment with JQ1 as indicated (**C**). In (**C**), respective values were normalized to the means of vehicle- (DMSO-) treated (JQ1 concentration zero) cultures. Asterisks indicate statistically significant changes [*, *p* < 0.05; **, *p* < 0.01; ***, *p* < 0.001; (**B**) top, middle: *t* test; *n* = 31/31 healthy/patient images from 3 preparations with 10–11 images per preparation and condition and 5–31 ROIs per image; bottom: *n* = 31/31 healthy/patient images from 3 preparations with 10–11 images per preparation and 3–29 ROIs per image; (**C**) One-way ANOVA with Tukey’s post hoc test; top, middle: 48 h: *n* = 31 to 33 images per concentration from 3 preparations with 10–12 images per treatment and 4 to 37 ROIs per image; 72 h: *n* = 32 to 34 images per concentration from 3 preparations with 10–13 images per treatment and 4 to 37 ROIs per image; bottom: 48 h: *n* = 31 to 33 images per concentration from 3 preparations with 10–12 images per treatment and 1 to 37 ROIs per image; 72 h: *n* = 32 to 33 images per concentration from 3 preparations with 8–13 images per treatment and 2 to 33 ROIs per image].

**Figure 4 ijms-26-05769-f004:**
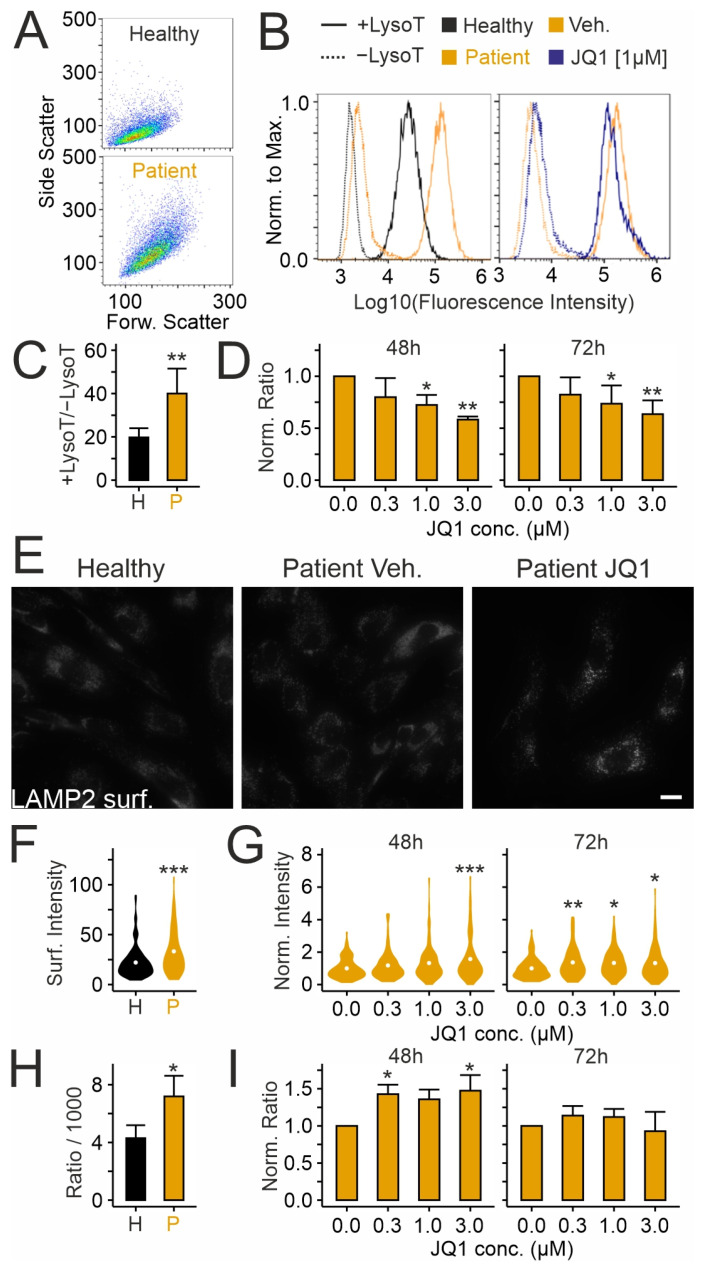
JQ1 reduces lysosomal expansion and induces lysosomal exocytosis in cultured skin fibroblasts from a NPCD patient: (**A**) Representative plots of side and forward scatter intensities of fibroblasts from a healthy donor (**top**) and from a NPCD patient (**bottom**) obtained by cytometry. (**B**) Histograms showing fluorescence intensities normalized to maximal counts in untreated cells from the healthy donor, and the NPCD patient treated with vehicle (DMSO) or with JQ1, and unstained (dotted line) or stained with lysotracker (solid line) as indicated. (**C**,**D**) Mean ratios of geometric means (lysotracker signal divided by background signal from cells without lysotracker) in untreated fibroblasts from the healthy (H) and the patient (P) donor (**C**), and in patient-derived fibroblasts following treatment with JQ1 (**D**) as indicated. In (**D**), ratios were normalized to the means of vehicle- (DMSO-) treated (JQ1 concentration zero) cultures. Asterisks indicate statistically significant changes [*, *p* < 0.05; **, *p* < 0.01; (**C**): *t* test; *n* = 4 preparations; (**D**): One-way ANOVA with Tukey’s post hoc test; *n* = 3–5 preparations]. (**E**) Fluorescence micrographs of cultured fibroblasts from a healthy donor (GM056549) and from an NPCD patient (GM18453) after treatment for 72 h with JQ1 or vehicle (DMSO). Following chemical fixation without permeabilization, cells were subjected to immunocytochemical staining to reveal the surface distribution of LAMP2. Scale bar: 20 µm. (**F**,**G**) Violin plots showing fluorescence intensities indicating surface expression of LAMP2 on somata of fibroblasts from the healthy (H) and the patient donor (P) without treatment (**F**), and in patient-derived fibroblasts following treatment with JQ1 as indicated (**G**). In (**G**), intensities were normalized to the mean fluorescence intensities in vehicle-treated (JQ1 concentration zero) cultures run in parallel. White points indicate means of normalized values. Asterisks indicate statistically significant changes [*, *p* < 0.05; **, *p* < 0.01; ***, *p* < 0.001; (**F**): *t* test; *n* = 153/131 healthy/patient-derived cells from 4 preparations; (**G**): One-way ANOVA with Tukey’s post hoc test; 48 h: 85–116; 72 h: *n* = 102–131 patient-derived cells per treatment from 3 to 4 preparations]. (**H**,**I**), Mean hexosaminidase activity normalized to cell numbers in untreated fibroblasts from the healthy (H) and the patient (P) donor, and in patient-derived fibroblasts following treatment with JQ1 as indicated (**I**). In (**I**), values (activity per cell number) were normalized to the means of vehicle-treated (JQ1 concentration zero) cultures. Asterisks indicate statistically significant changes [*, *p* < 0.05; (**H**): *t* test; *n* = 3–4 preparations; (**I**): One-way ANOVA with Tukey’s post hoc test; *n* = 3–4 preparations].

**Figure 5 ijms-26-05769-f005:**
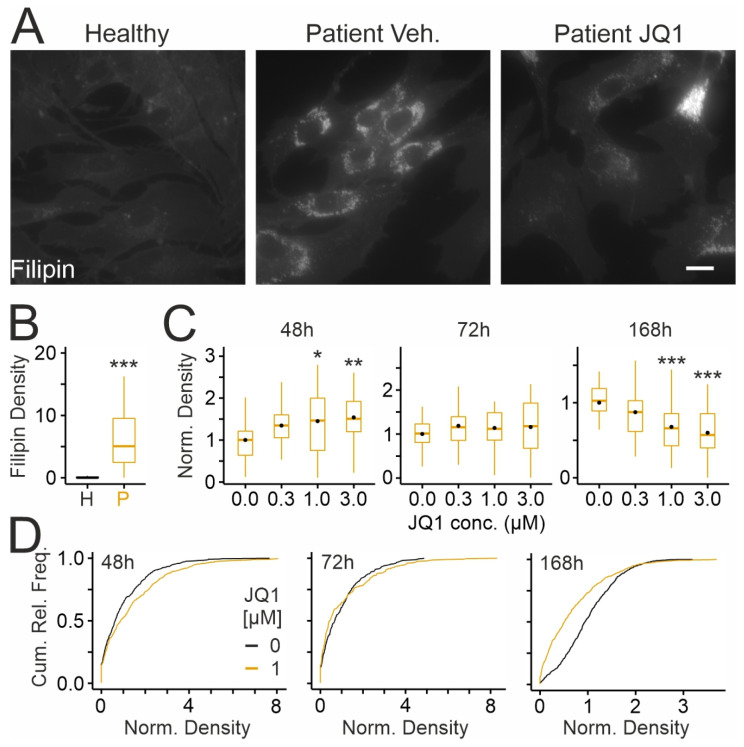
JQ1 affects cholesterol accumulation in cultured skin fibroblasts from an NPCD patient: (**A**) Fluorescence micrographs of cultured fibroblasts from a healthy donor (GM056549; **left**) and a NPCD patient (GM18453) following treatment for 72 h with vehicle (DMSO; middle) or with JQ1 (3 µM; **right**). After chemical fixation, cells were subjected to cytochemical staining with filipin to reveal the distribution of unesterified cholesterol. Scale bar: 20 µm. (**B**,**C**) Boxplots showing densities of filipin-positive puncta in fibroblasts from the healthy (H) and the patient (P) donor (**B**) and following treatment with JQ1 for indicated durations and concentrations (**C**). Black circles indicate mean values. Values in (**C**) were normalized to the means of vehicle-treated cultures. Asterisks indicate statistically significant changes [*, *p* < 0.05; **, *p* < 0.01; ***, *p* < 0.001; (**B**): *t* test; *n* = 45/45 healthy/patient images from 4 preparations with 10–11 images per preparation and condition and 6–30 ROIs per image; (**C**): One-way ANOVA with Tukey’s post hoc test; 48 h: 33–34 images per concentration from 3 preparations with 10–11 images per preparation and condition and 1–24 ROIs per image; 72 h: 33–34 images per concentration from 3 preparations with 10–12 images per preparation and condition and 6–27 ROIs per image; 168 h: 33–34 images per concentration from 3 preparations with 10–12 images per preparation and condition and 3–32 ROIs per image]. (**D**) Cumulative relative frequency plots showing densities of filipin-positive puncta in fibroblasts from the NPCD patient (GM18453) following treatment for indicated periods with JQ1 (1 µM) or with vehicle [DMSO; 48 h, vehicle: 450 ROIs/1 µM: 462; 72 h, vehicle: 527/434; 168 h: 818/519; same data as shown in panel (**C**)]. Note the time-dependent effects of JQ1 increasing and decreasing the densities at 48 h and 168 h, respectively, and the dual effect at 72 h with 30 and 70% of cells showing larger and smaller densities, respectively.

**Figure 6 ijms-26-05769-f006:**
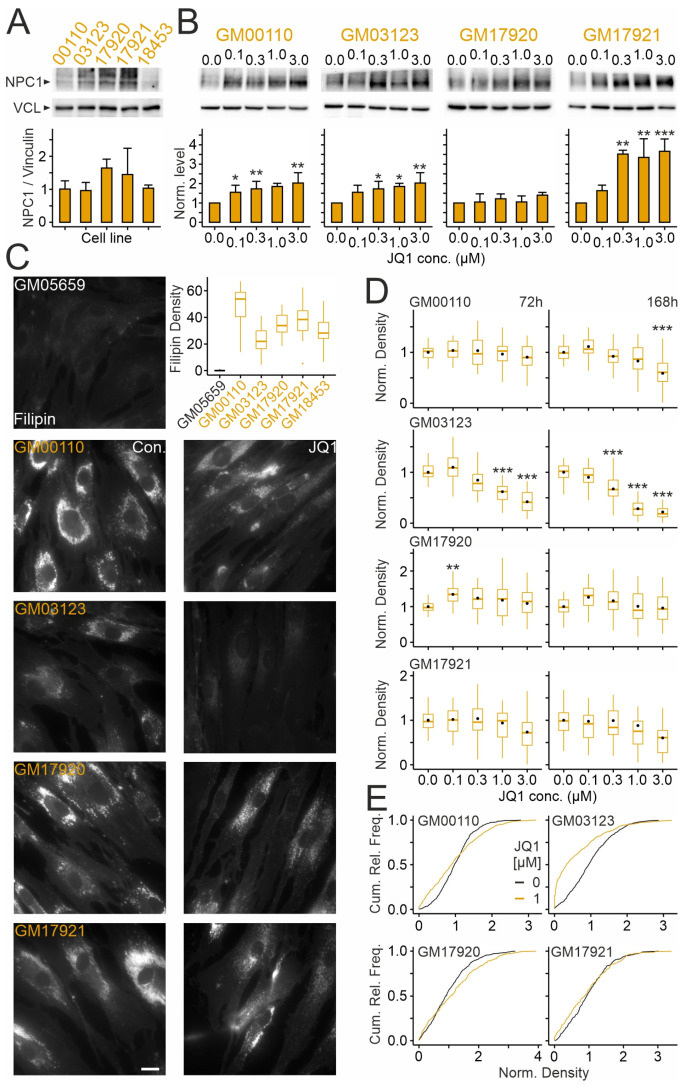
JQ1 enhances NPC1 levels and reduces cholesterol accumulation in cultured skin fibroblasts in a patient-specific manner: (**A**,**B**) Levels of NPC1 protein in primary cultures of skin fibroblasts from different NPCD patients under basal levels (**A**) and after treatment with JQ1 or vehicle (DMSO) at indicated concentrations for 72 h (**B**). Top, representative images of immunoblots showing bands corresponding to NPC1 and vinculin (VCL). Bottom, column plots in (**A**,**B**) showing mean values normalized to VCL levels [(**A**): *n* = 4 preparations] and to vehicle-treated (JQ1 concentration zero) control cultures [(**B**): *n* = 3–5 preparations], respectively. Asterisks in (**B**) indicate statistically significant changes compared to vehicle control (*, *p* < 0.05; **, *p* < 0.01; ***, *p* < 0.001; one-way ANOVA with Tukey’s post hoc test). (**C**) Fluorescence micrographs of cultured fibroblasts from a healthy donor and from different NPCD patients (indicated by codes) showing basal levels of unesterified cholesterol without (**left**; Con) and with JQ1 treatment (**right**; JQ1; 3 µM for 168 h). Cells were subjected to chemical fixation and cytochemical staining with filipin. Scale bar: 20 µm. Boxplots showing densities of filipin-positive puncta in somata of the different fibroblast lines under untreated conditions (GM05659: *n* = 502 ROIs; GM00110: *n* = 664; GM03123: *n* = 568; GM17920: *n* = 647; GM17921: *n* = 487; GM18453: *n* = 596; 41–45 images per line; 9–12 images per preparation; 4 preparations each). (**D**) Boxplots showing densities of filipin-positive puncta in fibroblasts from indicated patients following treatment with JQ1 at indicated durations and concentrations. Density values were normalized to the means of vehicle-treated (JQ1 concentration zero) cultures. Black circles indicate mean values. Asterisks indicate statistically significant changes (**, *p* < 0.01; ***, *p* < 0.001; one-way ANOVA with Tukey’s post hoc test; GM00110: 72 h: *n* = 481–659 ROIs, 32 images per conc., 8–34 ROIs per image; 168 h: 500–653 ROIs, 31–32 images per conc., 1–37 ROIs per image; GM03123: 72 h: 418–673 ROIs, 31–32 images, 8–33 ROIs per image; 168 h: *n* = 456–629, 28–32 images, 7–33 ROIs per image; GM17920: 72 h: *n* = 329–505 ROIs, 30 images, 3–23 ROIs per image; 168 h: *n* = 297–436 ROIs, 30 images, 4–21 ROIs per image; GM17921: 72 h: 263–338 ROIs, 30 images, 4–21 ROIs per image; 168 h: 249–354 ROIs, 29–30 images, 2–23 ROIs per image; 9–12 images per preparation, condition and line; *n* = 3 preparations per patient line). (**E**) Cumulative relative frequency plots showing densities of filipin-positive puncta in indicated fibroblast lines following treatment for 72 h with vehicle (DMSO) or JQ1 at 1 µM [subset of data shown in (**D**)]. Note that in nearly all lines, JQ1 induced dual effects by increasing and decreasing puncta densities.

**Figure 7 ijms-26-05769-f007:**
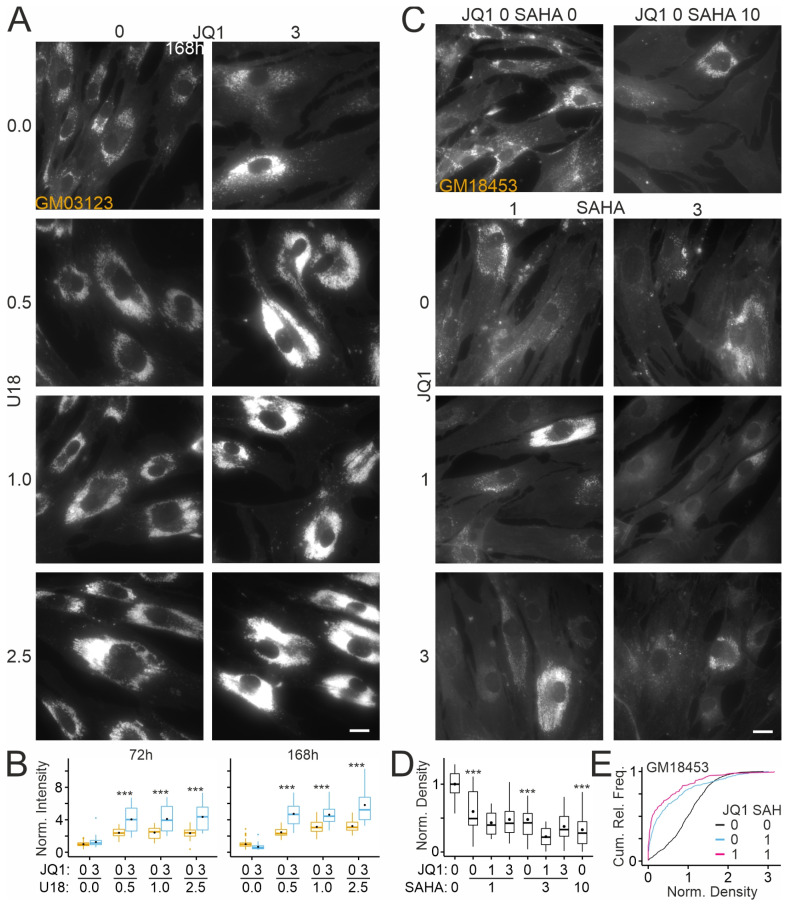
Effects of JQ1 on cholesterol accumulation in cultured skin fibroblasts in the presence of the NPC1 inhibitor U18 or of an HDAC inhibitor: (**A**) Fluorescence micrographs of cultured fibroblasts from a NPCD patient (GM03123) treated for 168 h with vehicle (DMSO/ethanol) or with U18 (microgram per mL) and JQ1 (micromolar) at indicated concentrations. Following treatments, cells were subjected to chemical fixation and cytochemical staining with filipin. Scale bar: 20 µm. (**B**) Boxplots showing fluorescence intensities of filipin in somata of fibroblasts treated for indicated durations with the indicated drug combinations. Intensity values were normalized to the means of vehicle-treated (JQ1 and U18 concentration zero) cultures. Black circles indicate mean values. Asterisks indicate statistically significant changes induced by JQ1 on U18-treated cells (***, *p* < 0.001; two-way ANOVA with Tukey’s post hoc test; 72 h: *n* = 406–818 ROIs, 37–40 images per condition, 2–92 ROIs per image; 168 h: 344–739 ROIs, 28–45 images per condition, 1–73 ROIs per image; 3 preparation). (**C**) Fluorescence micrographs of cultured fibroblasts from a NPCD patient (GM18453) treated for 72 h with indicated combinations of SAHA and JQ1 at micromolar concentrations. Following treatments, cells were subjected to chemical fixation and cytochemical staining with filipin. Scale bar: 20 µm. (**D**) Boxplots showing densities of filipin-positive puncta in somata of fibroblasts treated with the indicated drug combinations. Co-treatment with JQ1 and 10 µM SAHA was toxic to cells, precluding further analyses. Density values were normalized to the means of vehicle-treated (JQ1 and SAHA concentration zero) cultures. Black circles indicate mean values. Asterisks indicate statistically significant changes induced by SAHA compared to vehicle-treated cells (***, *p* < 0.001; two-way ANOVA with Tukey’s post hoc test; *n* = 368–605 ROIs, 29–40 images per conc., 6–29 ROIs per image). (**E**) Cumulative relative frequency plots showing densities of filipin-positive puncta in the patient-derived fibroblast line following indicated treatments for 72 h. Note the dual effect of SAHA, similar to that observed for JQ1 (Figure 5D and Figure 6E), and additive effects of SAHA and JQ1. Subsets of data are shown in panel (**D**).

## Data Availability

The raw data supporting the conclusions of this article will be made available by the authors on request.

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
