# Peer review of "Exploration of Bromodomain Proteins as Drug Targets for Niemann–Pick Type C Disease"

_ijms, 2025, doi:10.3390/ijms26125769_

Round 1
Reviewer 1 Report
Comments and Suggestions for Authors
In the present work, Parente and co-authors tested the effect of an inhibitor of BET proteins on the pathological phenotype of fibroblasts derived from NPC1 patients.
Some aspects of the paper are not clear and should be better addressed by the authors:
Line 69: the authors stated that the enhancement of lysosomal release is “transient” but LAMP2 surface expression is elevated at both 48 and 72h after JQ1 treatment (figure 4); did they refer to hexosaminidase activity shown in panel H that is increased only at 48h? The authors should better clarify this point.
Line 71: ”...in the presence of a cholesterol inhibitor” is not correct since U18 is considered an inhibitor on NPC1
Section 2.1: the authors found that the number of cells in patients-derived cultures are reduced over time, but this is not due to an increase in cell death, which was found significantly reduced; moreover, JQ1 inhibited cell growth in both NPC1 and healthy fibroblasts and significantly increased the number of dead cells; the authors should discuss this point, since the increase in cell death is usually considered a toxic effect of a drug.
The results shown in figures 2 and 3 seems in contradiction; in fact, after 72h of JQ1 treatment, NPC1 protein expression was found increased when evaluated by WB, while in immunofluorescence analysis such an increase is not observable. This discrepancy could be due to the representative fluorescence images used in the figure. Anyway, the authors should provide more representative images or should comment this discrepancy.
Section 2.2: the authors should briefly comment the results demonstrating that NPC1-fibroblasts display a basal higher level of lysosomal exocytosis than heathy ones.
Section 2.5: the results shown in figures 6D and 7B seem in contradiction; in fact, after 72h and 168h of treatment with 3microM JQ1, cholesterol accumulation is strongly reduced in GM03123 (figure 6D); in figure 7B, filipin intensity is not affected by JQ1 treatment at the same time points (first two histograms on the left of graphs at 72 and 168h, in absence of U18). How can the authors explain this discrepancy? Moreover, in my opinion this discrepancy could cause a mis-interpretation of the results: in fact, if the experiment aims at demonstrating that the reduction of cholesterol accumulation is mediated by the increase in NPC1 expression/activity induced by JQ1, I expect that JQ1 3microM, in absence of U18, causes a decrease in cholesterol accumulation (as it is expected to do on the basis of the results shown in figure 6D) and that the co-treatment with U18 reverts this reduction; thus, it could be concluded that the effect of JQ1 is NPC1-dependent (not -independent as stated by authors on lines 293-295). In addition, in the discussion session the authors stated that the effect of JQ1 is dependent on NPC1…this is very confusing. The authors should consider this point after justifying the above-mentioned contradictory results shown in Figures 6D and 7B.
In addition, as concerning Figure 6, representative micrographs of filipin intensity after JQ1 treatment should be shown for each of NPC1-fibroblast clones investigated, at least as a supplementary information.
Section 3, Discussion: on lines 338-339 and 349-351 the authors stated that “Our observation that JQ1 bolstered the effect of SAHA on cholesterol accumulation…”, but the results seem to not support this conclusion since statistical significance was not reached in Figure 7D; the sentence should be revised.
Minor concerns:
In section 4.9 and in some figure legends it is not specified which ANOVA test was used; this information should be added.
line 87: in the sentence “...dead cells fibroblasts in from...” “in” should be deleted.
line 112 (figure 2 legend): change “vinculin (VCL) concentration in “vinculin (VCL) expression”
line 376: change “apolipoprotein a” in apolipoprotein A”
Author Response
We thank both referees for the careful reading of out manuscript and their insightful and highly constructive comments.
Referee 1
In the present work, Parente and co-authors tested the effect of an inhibitor of BET proteins on the pathological phenotype of fibroblasts derived from NPC1 patients.
Some aspects of the paper are not clear and should be better addressed by the authors:
Comment 1: Line 69: the authors stated that the enhancement of lysosomal release is “transient” but LAMP2 surface expression is elevated at both 48 and 72h after JQ1 treatment (figure 4); did they refer to hexosaminidase activity shown in panel H that is increased only at 48h? The authors should better clarify this point.
Response 1: We thank the referee for pointing this out. We fully agree that the term "transient" did not properly reflect our findings, therefore we have removed the word in the Introduction.
Comment 2: Line 71: ”...in the presence of a cholesterol inhibitor” is not correct since U18 is considered an inhibitor on NPC1
Response 2: We have corrected this error and replaced "cholesterol" by "NPC1" (pg. 2, para 4).
Comment 3: Section 2.1: the authors found that the number of cells in patients-derived cultures are reduced over time, but this is not due to an increase in cell death, which was found significantly reduced; moreover, JQ1 inhibited cell growth in both NPC1 and healthy fibroblasts and significantly increased the number of dead cells; the authors should discuss this point, since the increase in cell death is usually considered a toxic effect of a drug.
Response 3: We agree with the referee that our results show toxicity of JQ1 as previous studies, notably after prolonged treatment and high concentrations, and that we did not state this clearly. We have modified the corresponding statements and cite relevant publications (pg. 2 last para.).
Comment 4: The results shown in figures 2 and 3 seems in contradiction; in fact, after 72h of JQ1 treatment, NPC1 protein expression was found increased when evaluated by WB, while in immunofluorescence analysis such an increase is not observable. This discrepancy could be due to the representative fluorescence images used in the figure. Anyway, the authors should provide more representative images or should comment this discrepancy.
Response 4: The referee raises an important point which we are happy to clarify. Indeed, our quantitative analyses reveal a discrepancy with respect to levels of NPC1 protein: we found a robust JQ-induced increase based on band intensities in Western blots but no change in density of fluorescent puncta or intensity correlation following immunocytochemical staining. We note that on the other hand, both approaches confirm the lower cellular level of the I1061T variant compared to the "normal" form thus indicating the validity of our approaches. The representative micrographs shown in Fig. 3 illustrate these differences. At present, it is unclear why the 1.5 fold increase in the I1061T variant of NPC1 observed by WB could not be detected by immunocytochemical staining. This may be due to the limited sensitivity of this method.
Comment 5: Section 2.2: the authors should briefly comment the results demonstrating that NPC1-fibroblasts display a basal higher level of lysosomal exocytosis than heathy ones.
Response 5: We thank the referee for raising a very important point. The difference in basal lysosomal exocytosis we found in our preparation has been observed previously. We have added a corresponding statement and citation in the Results (pg. 5, para. 2).
Comment 6: Section 2.5: the results shown in figures 6D and 7B seem in contradiction; in fact, after 72h and 168h of treatment with 3microM JQ1, cholesterol accumulation is strongly reduced in GM03123 (figure 6D); in figure 7B, filipin intensity is not affected by JQ1 treatment at the same time points (first two histograms on the left of graphs at 72 and 168h, in absence of U18). How can the authors explain this discrepancy?
Moreover, in my opinion this discrepancy could cause a mis-interpretation of the results: in fact, if the experiment aims at demonstrating that the reduction of cholesterol accumulation is mediated by the increase in NPC1 expression/activity induced by JQ1, I expect that JQ1 3microM, in absence of U18, causes a decrease in cholesterol accumulation (as it is expected to do on the basis of the results shown in figure 6D) and that the co-treatment with U18 reverts this reduction; thus, it could be concluded that the effect of JQ1 is NPC1-dependent (not -independent as stated by authors on lines 293-295). In addition, in the discussion session the authors stated that the effect of JQ1 is dependent on NPC1…this is very confusing. The authors should consider this point after justifying the above-mentioned contradictory results shown in Figures 6D and 7B.
Response 6: We thank the referee for the comment. It allows us to clarify important points. First, our statement on lines 293-295 was simply a typo error. We meant to say "dependent" as we stated in the Discussion. This is now corrected on pg. 10 para. 2. Second, with respect to the divergent effects of JQ1 on fibroblast line GM03123, we were as perplexed as the referee. Frankly, we do not know why in this particular series of experiments, effects of JQ1 +/- U18 (Fig. 7), were much less pronounced than the effects in the first series with this line (shown in Fig. 6). Third, with respect to U18 effects, we clarify our conclusions. In our opinion, the increase in filipin intensity in cells treated with U18 and JQ1 shows that there is residual activity of NPC1 I1061T as its blockade results in higher cholesterol accumulation. However, if JQ1 would act completely independently from NPC1, there would be no such increase. On the other hand, we cannot rule out the contribution of JQ1-independent pathways. They may be operational, but not sufficient to counteract the massive cholesterol accumulation following pharmacologic block of NPC1 with U18. We have added a corresponding statement on pg. 10 para. 2.
Comment 7: In addition, as concerning Figure 6, representative micrographs of filipin intensity after JQ1 treatment should be shown for each of NPC1-fibroblast clones investigated, at least as a supplementary information.
Response 7: As demanded by the referee we have included in Fig. 6C micrographs of filipin-stained fibroblasts following JQ1 treatment.
Comment 8: Section 3, Discussion: on lines 338-339 and 349-351 the authors stated that “Our observation that JQ1 bolstered the effect of SAHA on cholesterol accumulation…”, but the results seem to not support this conclusion since statistical significance was not reached in Figure 7D; the sentence should be revised.
Response 8: We agree with the referee, and we have revised corresponding statements in the Results (pg. 12, para. 2) and in the Discussion (pg. 12, para. 4).
Minor concerns:
Comment 9: In section 4.9 and in some figure legends it is not specified which ANOVA test was used; this information should be added.
Reponse 9: We have added the missing information.
Comment 10: line 87: in the sentence “...dead cells fibroblasts in from...” “in” should be deleted.
Reponse 10: Corrected.
Comment 11: line 112 (figure 2 legend): change “vinculin (VCL) concentration in “vinculin (VCL) expression”
Reponse 11: We changed to "levels of vinculin (VCL) protein".
Comment 12: line 376: change “apolipoprotein a” in apolipoprotein A”
Reponse 12: Corrected.
Reviewer 2 Report
Comments and Suggestions for Authors
The manuscript has a novel and promising epigenetic therapeutic strategy. Some suggestions to improve its structure and clarity:
1) Introduction can be more improved by clearly connecting BET proteins mechanistically with NPC1 regulation.
2) In method section, please clarify whether the mRNA level measured. Please clarify the exact normalization method in the lysosomal exocytosis essay. Please indicate the number of replicated of exocytosis essay in figure 4 as well as the control for non-specific staining.
3) Results: please add schematic explaining biphasic response overtime.
4) Discussion: please explain if BET inhibitor could be repurposed or tested in the future for NPCD clinical trials. Please highlight that the current drugs such as miglustat did not significantly lower the cholesterol in vitro which suggest the novelty of your findings. I would suggest adding summary figure or table in the discussion section.
Please consider checking the abbreviations when mentioned at first time (e.g., SAHA, U18). Please check figure caption formatting as some seem unclear without context.
Author Response
We thank both referees for the careful reading of out manuscript and their insightful and highly constructive comments.
The manuscript has a novel and promising epigenetic therapeutic strategy. Some suggestions to improve its structure and clarity:
Comment 1: 1) Introduction can be more improved by clearly connecting BET proteins mechanistically with NPC1 regulation.
Reponse 1: This is a very important point, and we have followed the referees suggestion to add possible corresponding explanations to the Introduction (see pg. 2 para. 3).
Comment 2: 2) In method section, please clarify whether the mRNA level measured.
Reponse 2: We have not measured mRNA levels and therefore, there is no paragraph related to this approach in the Methods section.
Comment 3: Please clarify the exact normalization method in the lysosomal exocytosis essay.
Reponse 3: We have used two assays, first immunocytochemical staining of non-permeabilized cells with LAMP2 antibody, second, measurements of hexosaminidase in culture medium. As indicated in the legend for Fig. 4G and 4I, the corresponding values (G: fluorescence intensities; I: enzyme activities per cell number) were normalized to mean values of corresponding vehicle-treated control cultures run in parallel. We have improved the description of the normalization method in the figure legend (pg. 6, para. 2).
Comment 4: Please indicate the number of replicated of exocytosis essay in figure 4 as well as the control for non-specific staining.
Reponse 4: The number of biological replicates (independent culture preparations) was indicated in the legend. Fig. 4F: 4 preps.; Fig. 4G: 3-4 preps.; Fig. H: 3-4 preps.; Fig. 4I: 3-4 preps.). We did not control for non-specific staining, since in Fig. 4F, we compare levels between fibroblasts from a healthy donor and a patient, and we have no evidence that there is difference between the different fibroblast lines in the "background signal" from immunocytochemical staining with secondary antibody only.
Comment 5: 3) Results: please add schematic explaining biphasic response overtime.
Reponse 5: We appreciate the referee's suggestion, but we prefer not to add a scheme in the Results part, as we do not really have an explanation for the "biphasic" response.
Comment 6: 4) Discussion: please explain if BET inhibitor could be repurposed or tested in the future for NPCD clinical trials.
Reponse 6: We appreciate the referee's encouraging and forward-looking remark, but in our opinion, it is too premature to refer to clinical trials based on the present data obtained in fibroblast-based experimental model, notably with respect to expectations of patient communities. The next obvious step are tests in models with specialized human cell types (e.g. neurons) and in NPCD animal models. We have added a statement at the end of the discussion to express our hopes (pg. 12, para. 2).
Comment 7: Please highlight that the current drugs such as miglustat did not significantly lower the cholesterol in vitro which suggest the novelty of your findings.
Reponse 7: We thank the referee for this excellent suggestion, we have added a corresponding statement and cite references (pg. 13 para. 1).
Comment 8: I would suggest adding summary figure or table in the discussion section.
Reponse 8: Again, we thank the referee for this suggestion. At this moment, we prefer to restrict ourselves to verbal description. We hope that our pioneering results lead to more studies on BET proteins in NPC disease, lysosomal function and cholesterol homeostasis. In the future, we may be able to draw schemes positioning BET proteins in these contexts.
Comment 9: Please consider checking the abbreviations when mentioned at first time (e.g., SAHA, U18).
Reponse 9: We have corrected this and explain now abbreviations at their first appearance.
Comment 10: Please check figure caption formatting as some seem unclear without context.
Reponse 10: We have removed the legend related to supplementary figure 1 from the main text.
Round 2
Reviewer 1 Report
Comments and Suggestions for Authors
The manuscript was changed according to reviewer's suggestions.
More changes are not necessary.